# Utilizing Immunocytokines for Cancer Therapy

**DOI:** 10.3390/antib10010010

**Published:** 2021-03-09

**Authors:** Erin Runbeck, Silvia Crescioli, Sophia N. Karagiannis, Sophie Papa

**Affiliations:** 1ImmunoEngineering Group, School of Cancer and Pharmaceutical Studies, King’s College London, London SE19RT, UK; runbecke@email.chop.edu; 2St. John’s Institute of Dermatology, School of Basic and Medical Biosciences, King’s College London, London SE1 9RT, UK; silvia.crescioli@kcl.ac.uk (S.C.); sophia.karagiannis@kcl.ac.uk (S.N.K.)

**Keywords:** antibody, antibody engineering, immunocytokine, immunotherapy, cytokines, cancer therapy

## Abstract

Cytokine therapy for cancer has indicated efficacy in certain diseases but is generally accompanied by severe toxicity. The field of antibody–cytokine fusion proteins (immunocytokines) arose to target these effector molecules to the tumor environment in order to expand the therapeutic window of cytokine therapy. Pre-clinical evidence has shown the increased efficacy and decreased toxicity of various immunocytokines when compared to their cognate unconjugated cytokine. These anti-tumor properties are markedly enhanced when combined with other treatments such as chemotherapy, radiotherapy, and checkpoint inhibitor antibodies. Clinical trials that have continued to explore the potential of these biologics for cancer therapy have been conducted. This review covers the in vitro, in vivo, and clinical evidence for the application of immunocytokines in immuno-oncology.

## 1. Introduction

For most of the 20th century, the role of the immune system in cancer was debated. It is now understood that the immune system is involved in the control of malignancy through ‘immunosurveillance’ [1]. Early studies indicated the importance of cytokines in this process as effector and co-stimulatory molecules. A report using fibrosarcoma cells expressing a dominant negative interferon-γ (IFN-γ) receptor, thus rendering the cells IFN-γ-insensitive in immunocompetent mice resulted in a loss of tumor control in the model [2]. Additionally, the discovery of interleukin 2 (IL-2) and its role in T cell expansion enabled ex vivo culture and expansion of T cells in a therapeutic context [3,4]. Interleukin 2 enhanced the cytotoxic potential of lymphocytes in two different syngeneic tumor models, thus prolonging subject survival, when lymphocytes were cultured with or injected simultaneously with IL-2 [5,6].

Tumor necrosis factor (TNF) has been approved for clinical use in Europe for isolated limb perfusions of sarcoma patients in conjunction with the chemotherapeutic agent melphalan. An initial trial of 23 patients treated with IFN-γ and TNF yielded an 89% complete response rate with high grade hematological toxicity in two patients [7]. A subsequent larger study had a similar high response rate of 153/186 patients resulting in limb salvage for previously unresectable tumors. Grade 4 regional adverse effects occurred in 14 patients, and grade 3–4 hepatic and cardiovascular toxicity occurred in 17 and 6 patients, respectively [8].

To date, only interferon-α (IFN-α) and IL-2 have been approved by the Food and Drug Administration (FDA) for cancer treatment [9]. Clinical trials showed that IFN-α2b had a response rate of 57/64 patients with hairy cell leukemia and 6/20 and 40/114 patients with Kaposi’s sarcoma in two separate studies [10,11,12]. The treatment of cutaneous melanoma with high dose IFN-α2b resulted in a median survival time of 3.82 years versus the control at 2.78 years [13]. A combination of chemotherapy and IFN-α2b in follicular lymphoma yielded an overall response rate of 104/123 patients compared to 82/119 patients in the chemotherapy-only arm [14]. However, the IFN-α therapy required a decrease in dosage or the cessation of treatment due to hematological, neurological, and hepatic toxicities [12,13,14].

Interleukin 2 administration resulted in objective response rates of 14% and 16% in metastatic renal cell carcinoma (mRCC) and metastatic melanoma patients, respectively [15,16]. Similar to IFN-α, IL-2 was found to have high rates of dose-limiting toxicities. Interleukin 2 treatment exhibited high grade cardiovascular, gastrointestinal, neurological, pulmonary, hepatic, renal, and hematological toxicities, and it led to deaths in 4% and 2% of mRCC and metastatic melanoma patients, respectively [15,16]. A trial combining systemic IL-2 with autologous ‘lymphokine activated killer’ (LAK) cells showed no significant increase in survival or responses in patients given both LAK cells and IL-2 compared to IL-2 alone. Additionally, IL-2 infusion was associated with high levels of severe toxicity, as highlighted by a 3.3% treatment related mortality rate [17]. These clinical trials for IL-2, TNF, and IFN-α illustrated the necessity to establish a less toxic method of cytokine therapy with a wider therapeutic window.

With the introduction of hybridoma technology [18], it became possible to create large libraries of monoclonal antibody-producing clones in order to specifically target antigens of interest. The evolution of recombinant antibody technology led to the phage display method that offers fully humanized antibodies independent of the host’s immune response [19,20,21,22]. Utilizing recombinant antibody production, payloads can be conjugated to antibodies and directed towards tumor-associated antigens (TAAs) for targeted therapy. As a method of targeted cytokine delivery, antibody–cytokine fusion molecules termed ‘immunocytokines’ (Figure 1) have been widely developed and assessed in clinical trials involving a variety of cancers and inflammatory conditions. They are composed of a targeting-antibody moiety, an amino acid linker, and a cytokine payload. Given the toxicity profiles of many systemically delivered cytokines, the use of an immunocytokine approach can often broaden the therapeutic window of the cytokine under study [23,24]. Initial reports showed the successful generation of immunocytokines containing lymphotoxin, TNF, granulocyte-macrophage colony-stimulating factor (GM-CSF), and IL-2 with direct and indirect anti-tumor capabilities in vitro [25,26,27,28,29]. An IL-2 immunocytokine targeting disialoganglioside (GD2) co-administered with LAK cells in a murine model for neuroblastoma resulted in the absence of detectable liver metastases. The injection of unconjugated recombinant human IL-2 (rhIL-2) at the same dosages as the immunocytokine was insufficient to abrogate liver metastases [30]. This field has since expanded to include different antibody formats, cytokines, combination regimens, and targeted indications. This review covers the evidence for immunocytokine therapy for the treatment of cancer.

Similar to monoclonal antibody therapy, the structure and size of the immunocytokine under investigation affects the performance of the drug. Full size antibody formats have increased circulation half-life and engage immune cells through the crystallizable fragment (Fc) portion of the molecule, retaining much of the antibody effector function within the immunocytokine [31,32]. However, immunocytokines utilizing antibody fragments also increase the half-life of the respective cytokine [24]. Smaller immunocytokine formats have a quicker clearance rate and greater tumor penetration [24,33]. Additionally, the structure of the cytokine payload can affect efficacy. An IL-12-based immunocytokine demonstrated an enhanced tumor retention and growth inhibition when expressed as a single chain variable fragment (scFv) heterodimer connected by a disulfide bridge between the two subunits of IL-12 compared to a monomer consisting of both IL-12 subunits fused by a linker [34]. The intended effect of the immunocytokine and the native conformation of the delivered cytokine should be taken into consideration during engineering.

By targeting the antibody portion to the tumor-associated antigens presented on tumor cells, the immunocytokine is sequestered in the tumor microenvironment. Here, the functional cytokine can signal through its cognate receptors expressed on either/both tumor and immune cells and induce an anti-tumor response. An upward-facing arrow denotes an increase in activity, while a downward-facing arrow denotes a decrease.

## 2. Immunocytokines in Pre-Clinical Models

Investigations have covered a multitude of different immunocytokine payloads and formats. In this section, previous reports of these antibody-fusion molecules will be addressed in the order of the delivered cytokine.

### 2.1. Interleukin-2

The most widely reported immunocytokines are IL-2-based. Interleukin-2 is a small protein primarily produced by activated T cells that increases proliferation and survival in an autocrine/paracrine fashion [35]. Based on the clinical evidence of unconjugated IL-2, targeted formats were developed to increase its therapeutic efficacy and safety. Early studies characterized the efficacy of IL-2 fused with an antibody specific for GD2 in models for melanoma and neuroblastoma [27,30,36,37,38,39,40]. In a murine model of metastatic melanoma, IL-2 immunocytokines targeting GD2 and epidermal growth factor receptor (EGFR) suppressed lung and liver metastases, in addition to prolonging survival, when administered after LAK cell reconstitution in immunocompromised mice. Unconjugated IL-2 and antibody injected after LAK cell reconstitution did not yield the same therapeutic effect [37]. In immunocompetent murine melanoma models, the immunocytokine anti-tumor response was dependent on CD8^+^ T cells and conveyed long-term protection against tumor rechallenge [36,38,39]. Later, it was determined that efficacy in pulmonary metastases models was also reliant on CD4^+^ T cell-dependent CD40/CD40 ligand signaling [41]. Immunocytokine efficacy was demonstrated in neuroblastoma and, in contrast to the melanoma model, was dependent on natural killer (NK) cells and not T cells [30,40,42]. This was attributed to the neuroblastoma cell line secreting factors such as transforming growth factor-β (TGF-β) and IL-10, which decreases T cell cytotoxicity, whereas select studies have shown IL-10 to increase NK cell activity in vivo [42,43]. Additionally, the intratumoral injection of GD2 or epithelial cell adhesion molecule (EpCAM)-targeting IL-2 immunocytokines increased the anti-tumor effect and immune cell presence in neuroblastoma and melanoma in vivo compared to intravenous administration and unconjugated IL-2 [44,45,46,47]. Both immunocytokines enhanced the formation of synapses between NK cells and melanoma or ovarian carcinoma, which was inhibited by an anti-IL-2 receptor α (IL-2Rα) antibody, suggesting that synapse formation was based on IL-2R engagement on NK cells by the cancer cell-bound immunocytokines [48].

Targets other than TAAs expressed on the cancer cells have been explored for IL-2 immunocytokine therapy. Immunocytokines have been designed to target the aberrant extracellular matrix (ECM) often found in the tumor micro-environment, which presents a more stable target in comparison to transformed cells. Different isoforms of fibronectin and tenascin-C are angiogenesis-related ECM proteins and show a tumor-vasculature restriction. Specifically, these vasculature TAAs are the extra-domain A (EDA) and extra-domain B (EDB) isoforms of fibronectin and the A1 domain isoform of tenascin-C (Tnc A1) [49,50,51,52]. An initial study investigating targeted IL-2 to EDB showed sufficient trafficking to the tumor vessels after intravenous administration and a significant reduction in tumor growth compared to saline, unconjugated IL-2, and parental antibody groups [23]. The delivery of IL-2 via EDA and Tnc A1 specificity showed efficacy at different dosages in acute myeloid leukemia (AML), non-small cell lung carcinoma (NSCLC), and teratocarcinoma models after intravenous or intratumoral injection; however, EDA-IL-2 was only effective when co-administered with an EDA-specific TNF immunocytokine [53,54,55]. Tumor control was dependent upon NK cell and CD8^+^ T cell presence, and immunocytokine treatment propagated an increase in CD8^+^ T cell proliferation and granzyme B secretion in the NSCLC tumors [53,54,55]. In the AML model, the immunocytokine caused a loss of weight in mice, but it was transient and did not require the cessation of treatment [54]. Interleukin-2 immunocytokines have been developed to target an array of antigens including prostate-specific membrane antigen (PSMA), programmed death ligand-1 (PD-L1), EGFR, carbonic anhydrase IX (CAIX), CD20, and DNA. As such, these immunocytokines increase the therapeutic window compared to unconjugated IL-2 and seem dependent on immune effector cells to illicit an anti-tumor response [31,56,57,58,59,60,61].

### 2.2. Interferons

Interferon-α, a type I interferon with multiple subtypes displaying similar effects, was initially characterized for its antiviral properties. In the context of cancer, it exhibits direct and indirect anti-tumor effects through the dysregulation of tumor-promoting genes, the upregulation of suppressor genes and major histocompatibility complex I (MHC I), the inhibition of cell replication, sensitization for cell death, and the stimulation of immune cells [62,63]. A CD20-targeting immunocytokine utilizing the IFN-α2 subtype decreased the 50% inhibitory concentration (IC_50_) compared to commercially available forms of IFN-α, increased antibody-dependent cellular cytotoxicity (ADCC) against lymphoma cells (with a decreased complement-dependent cytotoxicity (CDC)), and prolonged survival in murine groups treated with the immunocytokine compared to equal concentrations of parental IFN-α and antibody [63]. A similar immunocytokine-targeting human leukocyte antigen DR (HLA-DR) demonstrated an enhanced ability to stimulate the apoptosis-promoting signal transducer and activator of transcription 1 (STAT1) pathway in multiple myeloma and lymphoma cells compared to the CD20-specific immunocytokine, but it also mediated toxicity against healthy peripheral blood mononuclear cells (PBMCs) [64]. A bispecific hybrid of these two immunocytokines exhibited a greater cytotoxicity in Daudi lymphoma cells than either monospecific original immunocytokine and a similar toxicity to the monospecific anti-HLA-DR immunocytokine [65].

Interferon-α has also been targeted to checkpoint molecule PD-L1. The in vivo production of the T cell tumor trafficking chemokine C-X-C motif chemokine ligand 10 (CXCL10) increased in murine lung cancer cells accompanied by retarded tumor growth with anti-PD-L1-IFN-α compared to parental anti-PD-L1 and a non-specific immunocytokine [66]. Interferon-α immunocytokines targeting PD-L1 or CD20 demonstrated superior tumor control in lymphoma and colon cancer that was dependent on IFN-α receptor (IFNAR) expression in either the tumor cells or host immune cells [67,68]. In contrast, one group investigated an immunocytokine delivering IFN-α to the ECM protein EDA. It localized to the tumor and recruited immune cells but was no more effective at controlling the growth of teratocarcinoma or melanoma xenografts than a non-specific immunocytokine control. One possible explanation for this lack of efficacy is that increased IFN-α located in the perivasculature may deliver less of an anti-tumor response than if directly targeted to the tumor cells [69].

Interferon-γ is a type II interferon produced by activated T cells and NK cells. It has anti-viral/anti-tumor characteristics and stimulates innate and adaptive immune responses [70]. Early studies exemplified the importance of IFN-γ for inherent protection against tumor formation and progression. Investigations using transgenic mice lacking the crucial cell-mediated cytotoxicity molecules or receptors for IFN-γ and perforin led to a higher rate of chemically-induced tumor development, the increased tumor growth of syngeneic xenografts, and an elevated proportion of metastases [71,72]. Targeting IFN-γ to the tumor microenvironment via EDA or EDB resulted in enhanced tumor and liver metastasis control when compared to a non-specific IFN-γ immunocytokine in teratocarcinoma models, sarcoma, and Lewis lung carcinoma (LLC) models. Increased numbers of CD4^+^ T cells, NK cells, macrophages, and granulocytes were observed in the tumors treated with targeted IFN-γ with a decrease in regulatory T cells (Tregs). Both immunocytokines exhibited impeded tumor trafficking by the IFN-γ receptor binding off-target tissues. Tumor specificity could be potentiated by saturating receptors with additional immunocytokine administration and no apparent effect on body weight [73,74].

Smaller antibody formats, such as single-domain variable heavy chain (VHH) antibodies, with potentially greater tumor penetration, and more tumor restricted interferons such as IFN-λ have also been explored in the context of immunocytokine therapy [61,75].

### 2.3. Tumor Necrosis Factor

The earliest reports of antibody–cytokine fusion molecules included TNF. Tumor necrosis factor is a pro-inflammatory cytokine expressed by macrophages, NK cells, T cells, and B cells. While studies have shown the anti-tumor effects of TNF administration, endogenous production can also promote tumor cell survival through its activation of transcription factor NF-κB and is expressed by various cancer cells types [76,77,78,79,80]. Due to its inflammatory potency, TNF is a good prospect for localized therapy and was initially targeted to the human transferrin receptor and GD2 with cytotoxicity in vitro in breast cancer cells and murine fibroblasts [25,81,82]. The tumor necrosis factor delivery to tumor microenvironment antigens such as EDA, EDB, and fibroblast activation protein (FAP) demonstrated growth inhibition in multiple models. Anti-FAP-TNF stimulated polymorphonuclear (PMN) cells to generate reactive oxygen species and human umbilical vein endothelial cells (HUVECs) to produce the clotting molecule tissue factor, which can cause necrosis and granulocyte infiltration, while anti-EDA-induced necrosis and NK cell infiltration [53,83]. Targeting TNF to EDB delayed tumor growth in teratocarcinoma models compared to untargeted cytokine, and the anti-cancer effect was amplified when co-administered with melphalan or an EDB-specific IL-12 immunocytokine [24,84]. Necrosis was seen when TNF was targeted to the described hypoxic marker CAIX along with growth inhibition and protection against rechallenge in CAIX^+^ colon and renal cell carcinoma-bearing mice with q 3% decrease in body weight [59,85]. Tumor necrosis factor-resistance in a breast cancer cell line was overcome by a TNF immunocytokine targeting human epidermal growth factor 2 (HER-2) via the upregulation of TNF receptor 1 (TNFR1) expression, the activation of caspase-associated apoptosis, and the deactivation the Akt proliferation pathway [86,87]. Overall, TNF immunocytokines engage anti-tumor immunity through direct cytotoxicity and the induction of necrosis of tumor tissue.

### 2.4. Interleukin-12

Interleukin 12 is a heterodimeric cytokine produced by most leukocytes and skews immunity towards an anti-cancer T helper cells type 1 (T_H_1) effector response [88]. High systemic toxicity rates at low dosages of IL-12 led to its development for targeted delivery [89,90]. An early study of targeted IL-12 to EDB showed tumor growth control in colon carcinoma and, to a greater extent, teratocarcinoma models. This was accompanied by a 6% body weight loss and pathological findings in harvested liver tissue. There was an increased IFN-γ and CXCL10 production in the tumor environment and an influx of CD8^+^, CD4^+^ T, and NK cells [90]. The modification of this particular immunocytokine’s format indicated that a heterodimeric fusion protein containing two anti-EDB scFvs, linked by a natural disulfide bridge in the two subunits of IL-12, had greater tumor trafficking, retention, and efficacy in teratocarcinoma than the previous single scFv-IL-12 format [34]. Variability has also been observed with different linker lengths and sequences attaching IL-12 and IL-2, with specific linkers enhancing tumor localization and retention [91,92]. Targeting IL-12 to EDB and EDA has been efficacious in tumor models of teratocarcinoma, prostate cancer, colon cancer, and epidermoid carcinoma with an influx of inflammatory innate and adaptive leukocytes [93,94,95].

A well-described IL-12 immunocytokine includes an anti-DNA fragment-targeting portion in order to direct the cytokine to necrotic tumor tissue. The immunocytokine outperformed unconjugated IL-12 in delaying the progression of LLC, colon carcinoma, melanoma, and bladder cancer tumors. This anti-cancer response and the prolongation of survival was reliant on NK cells and CD8^+^ T cells, but not CD4^+^ T cells, and ultimately did not protect against rechallenge [96,97]. An analysis of the microenvironment showed a shift towards a more inflammatory cellular response comprised of an increase in the CD8/CD4 T cells:macrophage/myeloid-derived suppressor cell (MDSC) ratio, a decrease in intratumoral TGF-β, and an increase in the proliferation/activation of CD4^+^ and CD8^+^ T cells [97]. This immunocytokine has been trialed in dogs with malignant melanoma with 2/15 partial responses and four experiencing grade 4–5 toxicity [98].

### 2.5. Interleukin-15

Similar to IL-2, IL-15 induces proliferation in T cells. In contrast, IL-15 preferentially promotes the proliferation of cytotoxic T cells rather than Tregs and decreases activation-induced cell death (AICD) compared to IL-2. Furthermore, IL-15 promotes the differentiation and maintenance of memory T cells and NK cells [99]. Targeting IL-15 to EDB indicated stronger efficacy in teratocarcinoma and colon carcinoma models compared to non-specific immunocytokine in a CD8^+^ T cell-dependent manner [100]. Interleukin-15 is primarily effective through trans-presentation: it is bound to IL-15 receptor α (IL-15Rα) on the surface of monocytes, macrophages, and dendritic cells, which then bind to IL-15Rβ/γc on T cells and NK cells [101]. Because of this, most immunocytokine studies based on IL-15 have developed a recombinant cytokine, RLI, that encompasses IL-15 bound to a portion of the IL-15Rα called the sushi domain. The trans-presentation immunocytokine caused the greater proliferation of PBMCs in vitro compared to anti-FAP-IL-15 and increased the cytotoxicity of T cells [102]. Targeting RLI to GD2 and CD20 indicated ADCC and CDC in neuroblastoma and lymphoma cells, reduced the number of liver metastases, and prolonged the survival of mice for up to 120 days in lymphoma models compared to parental proteins [103,104].

### 2.6. Granulocyte-Macrophage Colony-Stimulating Factor

Granulocyte-macrophage colony-stimulating factor is an inflammatory cytokine involved in the maintenance and maturation of myeloid cells, and it stimulates dendritic cell and macrophage activity [105]. This cytokine has been employed for immunotherapeutic regimens such as the FDA-approved dendritic cell vaccine for prostate cancer, Provenge [106]. Early studies described GM-CSF immunocytokines with varying activity [82,107]. An anti-GD2 antibody conjugated to GM-CSF showed increased neuroblastoma cell lysis compared to an unconjugated antibody when incubated with PMN cells, and cell death was attributed to ADCC [108]. Targeting GM-CSF to HER-2 reduced colon carcinoma tumor growth in a syngeneic model, initiated a T_H_2-type antibody response, and increased the antigen presentation of dendritic cells [109,110]. When targeting EDB, the immunocytokine was able to traffic to teratocarcinoma tumors and delay progression at repeated high doses compared with a non-specific immunocytokine [100]. The dual delivery of IL-2 and GM-CSF to EpCAM-expressing colon carcinoma in vivo resulted in less lung metastases at low doses, but the effect of the dual immunocytokine was not greater than the single cytokine-carrying molecules [111].

### 2.7. Additional Cytokines

Various cytokine payloads have been investigated, with the most common listed previously. Additional studies have described cancer immunocytokine therapy incorporating tissue factor, IL-4, IL-3, granulocyte colony-stimulating factor (G-CSF), IL-1β, IL-6, IL-13, IL-17, IL-7, vascular endothelial growth factor (VEGF), C-C motif chemokine ligand 21 (CCL21), and 4-1BBL [95,112,113,114,115,116,117,118,119,120,121,122,123,124].

### 2.8. Variants

Because of the potency of some inflammatory cytokines and the toxicity reported in trials, an added measure to increase safety and efficacy in immunocytokines is the use of variant cytokines. A point mutation in the IL-2 sequence responsible for binding endothelial cells, and thus off-target toxicity, resulted in a modest decrease of the binding of the high affinity receptor (IL-2Rα) expressed on T cells and a dramatically decreased binding of the intermediate affinity receptor expressed on other tissues. This led to preferential proliferation in T cells over NK cells and a 20-fold reduction in toxicity compared to the wild type [125]. In contrast, a carcinoembryonic antigen (CEA)-targeting IL-2 variant with diminished IL-2Rα binding, and thus preferential IL-2Rβ association that is predominant on cytotoxic T cells, was developed to reduce Treg response and off-target receptor binding. Proliferation in Tregs was greatly reduced in the IL-2 mutant compared to the wild type, with no difference in CD8^+^ T cell or NK cell proliferation and an increased tumor uptake [126]. The same group also fused this IL-2 variant to a PD-1-targeting antibody to directly deliver IL-2 to effector cells. The administration of the anti-PD-1 IL-2 variant eradicated pancreatic tumors in orthotopic models, while the co-administration of anti-PD-1 and an irrelevant IL-2 variant immunocytokine did not [127]. Another IL-2 variant with decreased binding to high affinity IL-2Rα but increased binding to intermediate affinity IL-2Rβ showed the augmentation of intratumoral CD8/Treg ratio and efficacy in melanoma and colon carcinoma models [128].

Variations have been developed without mutating the IL-2 sequence itself. One such variation includes the linkage of the chemotherapeutic drug DM1. This drug-conjugated immunocytokine retained IL-2 signaling in T cells and tumor-homing capabilities, but it exhibited a significant increase in tumor control with no elevation of toxicity [129]. An alternative structure design of cytokine conjugation to the light chain instead of the heavy chain of a full length antibody increased the half-life and therapeutic effect through the hinderance of the IL-2Rβ-binding portion of IL-2 [130]. Without specifically targeting a TAA, one group developed an IL-2/anti-IL-2 antibody complex that blocked IL-2 association with IL-2Rα. This antibody complex increased the cytokine’s half-life in circulation, preferentially expanded CD8^+^ T cells over Tregs, and augmented the anti-tumor response [131,132]. Similarly, the specific PEGylation of unmutated IL-2 diminished IL-2Rα binding and biased the molecule towards IL-2Rβ. This PEGylated IL-2 exhibited greater tumor retention and circulation half-life, increased infiltrating CD8^+^ T cells, and had superior tumor growth inhibition compared to unconjugated IL-2 in multiple murine models [133,134]. Tumor inhibition was significantly increased when co-administered with checkpoint inhibitors anti-CTLA-4 or anti-PD-1 [133,135]. This selective IL-2 variant has since progressed to phase II and III clinical trials in combination with anti-PD-1 (nivolumab) for advanced solid tumors, demonstrating a 50% overall response rate in PD-L1^+^ tumors [136,137,138,139].

In an attempt to increase the safety profile of IFN-α, a mutant was developed with a greater dissociation rate from its receptor, and it resulted in enhanced tumor-specific lysis while sparing healthy cells compared to the wild type [140,141]. This design led to tumor accumulation and efficacy in lymphoma, multiple myeloma, and melanoma models comparable to untargeted and wild-type IFN-α immunocytokines, though with significantly less toxicity [141,142]. A similar concept incorporated a mutated TNF or IFN-γ with decreased affinity for its receptor and showed target-specific cytotoxicity, induced cellular adhesion molecules in the tumor, and did not increase toxicity [143]. A TNF mutant with a reduced potency was generated for an IL-2-TNF dual immunocytokine and controlled tumor progression when targeted to EDA in colon carcinoma, LLC, sarcoma, and teratocarcinoma models [144]. A strategy to limit the off-tumor effects of IL-12 involved separate constructs encoding the p35 and p40 subunits targeting EDA for dimerization/activation in the tumor microenvironment. However, this study suggested that the p35 subunit immunocytokine retained IL-12 signaling capabilities in T cells and NK cells [145].

## 3. Immunocytokines in Combination Therapeutic Approaches

Combinatorial regimens with immunocytokines have suggested additive or synergistic effects in pre-clinical tumor models compared to monotherapies. These investigated combinations include immunocytokines with chemotherapy agents, radiotherapy, monoclonal antibodies, cytokines, small molecule inhibitors, and vaccines.

### 3.1. Chemotherapy

As chemotherapy is the gold standard treatment for most advanced malignancies, the addition of immunocytokines have been explored in order to increase the efficacy of already established therapies. An early study indicated a significantly enhanced tumor growth control when administered in conjunction with paclitaxel or cyclophosphamide, at inefficacious dosages alone, in syngeneic breast, colon, and lung carcinoma models [146]. Similar results were seen in the context of targeting the tumor microenvironment with IL-2-based immunocytokines. In a xenograft model of melanoma, the EDA-specific immunocytokine abolished tumors when injected 4 or 24 h after paclitaxel. A look into the mechanism showed an increased vascular perfusion, immunocytokine tumor-homing, and NK cells in the tumor of combination-treated mice [147]. In a syngeneic melanoma model, the same EDA-targeting IL-2 immunocytokine increased the efficacy of both paclitaxel and dacarbazine [148]. Utilizing an IL-2 immunocytokine-targeting Tnc A1 augmented the control of tumor progression for both paclitaxel and doxorubicin in breast cancer-bearing mice without additional toxicity [149]. Targeting IL-2 to Tnc A1 in combination with temozolomide exhibited an increase in the arrest of tumor progression in subcutaneous and intracranial glioblastoma models compared to monotherapy. The combination significantly increased tumoral apoptosis and the infiltration of NK cells and macrophages [150].

Tumor necrosis factor-based immunocytokines have also shown an augmented therapeutic effect when combined with different chemotherapeutic drugs. A combination of HER2-targeted TNF with 5-fluorouracil (5-FU) significantly inhibited the growth of pancreatic cancer cells compared to monotherapy of either agent. This combination downregulated anti-apoptotic molecule Bcl-2 and signaling through the survival Akt pathway while upregulating the TNF receptor [151]. Targeting a trimeric TNF to EDA led to cures in two different sarcoma models when given in conjunction with doxorubicin at a concentration insufficient to cause tumor regression alone [152]. A TNF immunocytokine specific for EDB reiterated these results by increasing the cure rate for dacarbazine and trabectedin in a syngeneic sarcoma model with a temporary acute loss of body weight [153]. Other studies have demonstrated the increased anti-tumor efficacy of combination chemotherapy and IL-12, IL-7, IFN-γ, and IFN-α immunocytokines [73,96,123,142,154].

### 3.2. Radiotherapy

Radiotherapy is used for the treatment of cancer because of its localized cytotoxicity and evident stimulation of an anti-cancer immune response [155]. Because of its immunostimulatory nature as well, the effects of IL-2 immunocytokines combined with radiation therapy have been documented. External beam radiation alone delayed tumor progression, but in combination with the intratumoral delivery of a GD2-targeting IL-2 immunocytokine, there was significant tumor regression and survival in melanoma models. This reaction was dependent on ADCC and T cell infiltration, and it conferred protection against rechallenge. Even greater anti-tumor responses were seen when radiation, immunocytokine, and anti-CTLA-4 blockade were combined in immunocytokine antigen-positive tumors [156,157]. However, the presence of a secondary untreated tumor negatively impacted the immunocytokine/radiation efficacy in primary treated pancreatic and melanoma tumors. This therapeutic inhibition was negated upon Treg depletion, anti-CTLA-4 addition, or the irradiation of all tumors [158]. In contrast, anti-EDB-IL-2 with radiation in colon cancer models caused a systemic anti-tumor response with control of tumor progression seen in secondary non-irradiated tumors in the combination group. This combination increased CD8^+^ T cells with a memory-like phenotype and protected against tumor rechallenge [159]. Radiotherapy given prior to anti-EDB-IL-2 significantly inhibited tumor progression in teratocarcinoma compared to monotherapy, with higher levels of infiltrating NK cells, and showed antigen density-dependency in colon, breast, and Lewis lung carcinoma [160,161]. A classified ‘cold’ neuroblastoma tumor achieved complete tumor eradication upon immune stimulation with anti-CTLA-4, anti-CD40, and toll-like receptor 9 (TLR9) agonist in addition to radiation and anti-GD2-IL-2 [162]. Additional studies have shown increased tumor control in vivo with combinations of radiotherapy and IL-2 immunocytokines targeting CEA and EpCAM [163,164,165].

Targeting IL-12 to the tumor in conjunction with radiation has indicated an increased benefit. Additive therapeutic efficacy was observed with the combination of radiation and a DNA-targeting IL-12 immunocytokine compared to monotherapies in syngeneic LLC models, along with enhanced tumor localization, after radiation was administered in xenograft rhabdomyosarcoma models [96,166]. Mice treated with radiation and DNA-targeting IL-12 showed a decreased tumor burden for both irradiated and non-irradiated tumors. This was accompanied by an increase in effector cells and a shift in the tumor microenvironment to a more pro-inflammatory cytokine milieu [167].

### 3.3. Monoclonal Antibodies

Monoclonal antibodies are the immunotherapeutic class with the greatest number of FDA approvals to date. The anti-CD20 monoclonal antibody rituximab was developed by IDEC pharmaceuticals and became the first FDA-approved cancer therapeutic monoclonal antibody [168]. This was followed by a cascade of FDA approvals for therapeutic antibody treatment in multiple blood cancers and solid tumors. A combination of rituximab and EDB-specific IL-2 immunocytokine eradicated tumors in multiple lymphoma xenografts with no apparent toxicity, while rituximab combined with IL-2 only delayed progression [169,170]. Immunocytokine alone, and to a greater extent with rituximab, increased macrophage and NK cell presence in the tumor compared to controls [170]. An antibody targeting syndecan-1 inhibited vascular tubule formation, and when co-administered with this EDB-targeting immunocytokine, melanoma tumors were ablated in ~71% of mice and a significant reduction occurred in ovarian tumor progression [171,172]. Targeted TNF to the melanoma antigen gp75 in combination with a monoclonal antibody of the same specificity augmented the efficacy in syngeneic melanoma models compared to either therapy alone. There was a loss of no more than 10% body weight in groups treated with immunocytokines [173].

While initial antibody development was targeted against tumor-associated molecules, there was a paradigm shift towards the development of antibodies that instead disrupted natural immune checkpoints that would otherwise suppress immune responses. The application of checkpoint inhibitors (CPIs) has led to the largest class of approved antibodies to date with the most indications in advanced solid tumors. Thus far, monoclonal antibody therapies targeting the immune checkpoints CTLA-4, PD-1, and PD-L1 have become commercially available, and so combinations with various immunocytokines have been investigated to enhance anti-cancer potential. Checkpoint inhibitors antagonistic for PD-L1 have exhibited increased survival and tumor control when combined with a decreased affinity IL-2-variant immunocytokine that targets CEA. The immunocytokine also enhanced the therapeutic efficacy of anti-HER2 and anti-EGFR [126]. A similar IL-2 mutant immunocytokine targeting EGFR had a synergistic effect with anti-PD-L1 in melanoma models [128]. In the context of a DNA-specific IL-12 immunocytokine, combination with anti-PD-L1 delayed colon, orthotopic bladder, and breast carcinoma progression with protection against rechallenge [174,175]. The immunocytokine with anti-PD-L1 had higher circulating levels of inflammatory cytokines, effector memory T cells, and intratumoral macrophages with high MHC expression compared with anti-PD-L1 alone [175]. Approximately 80% of tumor eradication was seen in colon cancer and sarcoma models when anti-PD-L1 was co-administered with IL-2 or TNF and IL-2 immunocytokines targeted to EDA and EDB [144,153]. The CPI anti-CTLA-4 has been studied with immunocytokine therapy and an anti-GD2-IL-2 immunocytokine increased the efficacy of CTLA-4 CPI in melanoma models in a T cell-dependent manner. Established tumor regression occurred with the further addition of innate immune cell activators anti-CD40 and TLR9 agonist [176]. Anti-CTLA-4 combined with EDB-targeted IL-2 caused teratocarcinoma and colon tumor regression to a greater extent when the immunocytokine was injected first [177].

### 3.4. Other Combinations

Immunocytokine therapy has had successful results upon incorporation into other established therapies. Various immunocytokines have been combined with additional unconjugated cytokines such as IL-7, IL-2, Fms-like tyrosine kinase-3-ligand (Flt3-L), and IFN-γ to show enhanced tumor control and the activation of the immune response [85,178,179,180]. Of course, to increase the specificity and thus safety of cytokine addition, immunocytokines have been investigated in varying combinations with each other, including targeted TNF, IL-8, IFN-γ, IL-2, and IL-4 [73,74,95,148,177,181,182]. Immunocytokines have shown successful potentiation with other therapies such as small molecule inhibitors, vaccines, oncolytic viruses, resveratrol, and adoptive cell therapy [96,128,183,184,185,186,187,188,189,190,191,192,193,194].

## 4. Clinical Investigations of Immunocytokines

Based on the pre-clinical evidence, many immunocytokines have progressed to phase I/II clinical trials for different indications (Table 1). One of the first immunocytokines to make it to clinical trials delivered IL-2 to the GD2 antigen commonly found in melanoma and neuroblastoma tumors. A pilot study in melanoma patients showed dose-limiting toxicities of hypoxia, thrombocytopenia, hypotension, and hyperglycemia in 7/33 patients, all of which were reversible [195]. A phase I trial of this immunocytokine in pediatric neuroblastoma and one melanoma patient indicated tolerability with 8/28 dose-limiting toxicities including neutropenia, leukopenia, hematuria, hypotension, thrombocytopenia, blurred vision, allergic reaction, and rash. Immune stimulation was observed with soluble circulating IL-2R and peaked at day four of treatment [196]. The further analysis of immune activation in metastatic melanoma patients showed increased T cell infiltrates in the tumor after treatment and a decrease in the targeted antigen GD2 [197]. Phase II trials in metastatic melanoma and neuroblastoma patients yielded 1/14 partial responses and 5/23 complete responses, respectively [198,199]. To optimize the therapy in melanoma, a recent pilot trial in advanced resectable melanoma patients with a lower tumor burden resulted in a median recurrence-free survival (RFS) of 5.73 months but 6/18 RFS cases at ~57 months [200]. The optimization of this immunocytokine in neuroblastoma led to a phase II trial in combination with GM-CSF and the vitamin A derivative isotretinoin. Five out of 45 patients had objective responses, and three of those had no events for up to five years. Four patients had dose-limiting toxicities [201]. The evaluation of the immune response to therapy showed that 32/61 patients developed anti-immunocytokine idiotype antibodies, which were not correlated with toxicity, and responses in neuroblastoma were associated with killer immunoglobulin-like receptor (KIR)/KIR ligand mismatch in NK cell activity [202,203]. In the context of resected advanced melanoma, patients treated with immunocytokine before resection showed a correlation between RFS/overall survival and inflammatory immune cell infiltration [204].

EDB-specific immunocytokines with payloads of TNF, IL-2, or IL-12 have been studied in phase I and II clinical trials. A first-in-man trial delivering TNF to solid tumors had only grade 1–2 adverse events, except one patient with grade 3 bone pain at a site of metastasis, and stable disease in 19/31 participants [213]. A more localized approach evaluated the intratumoral administration of the IL-2 EDB-targeting immunocytokine in 24 stage III melanoma patients and was well-tolerated, with complete responses seen in 44% of treated lesions and 45% of non-treated lesions. A significant increase was seen in circulating CD4^+^ T cells, CD8^+^ T cells, NK cells, and Tregs, and a decrease was seen in circulating MDSCs [214]. A phase II trial incorporating anti-EDB IL-2 and TNF immunocytokines in 20 stage III and IV melanoma patients resulted in 30% of lesions experiencing a complete response. There was increased necrosis in biopsies from the treated lesions and a significant infiltration of CD8^+^ T cells [215]. This combination of EDB-targeting immunocytokines has since progressed to phase III trials for intratumoral delivery to melanoma patients prior to resection (NCT02938299 and NCT03567889). A phase I study for the combination of the IL-2 immunocytokine and the chemotherapeutic agent dacarbazine in metastatic melanoma exhibited low rates of severe toxicity, mainly grade 4 leukopenia/neutropenia and grade 3 hypotension. Of the 29 patients, 28% had an objective response to the therapy [216]. This combination progressed to a phase II trial in which 7/38 patients receiving both therapies experienced an objective response compared to 1/22 patients receiving only dacarbazine. The mean progression-free survival (PFS) for the two differentially scheduled combination treatment groups was 99 days and 74 days compared to 59 days for the control group, although this was not statistically significant. Four cases of grade 4 neutropenia and one case of grade 4 thrombocytopenia were reported in the combination groups [217]. Further targeting IL-2 to EDB in solid tumors and RCC in a phase I/II study indicated tolerable toxicity with no grade 4 events and most of the grade 3 events occurring at the higher dose. Seventeen out of 33 patients presented with stable disease, and the mean PFS in the phase II arm was eight months, with two patients surpassing 24 months [218]. An immunocytokine utilizing a different EDB-specific antibody conjugated to IL-12 was trialed in melanoma and RCC patients and exhibited tolerability with transient grade 3 hematological toxicity, fatigue, fainting, dehydration, and urinary frequency events. The immunocytokine increased the circulating levels of IFN-g and CXCL10. Stable disease was experienced in 5/13 patients, with one partial response [219]. A comprehensive list of clinical trials involving immunocytokines for the treatment of cancer is detailed in Table 1.

## 5. Summary

Pre-clinical evidence is strong for targeted cytokine delivery through the immunocytokine format in different indications and has led to multiple clinical trials. Few immunocytokines have progressed to phase II trials, and only two (anti-EDB-IL-2 and anti-EDB-TNF) have progressed to a larger scale phase III. Continual investigation into the optimization of this immunotherapy with combinatorial agents could bring to light an encompassing understanding of its therapeutic value.

## Figures and Tables

**Figure 1 antibodies-10-00010-f001:**
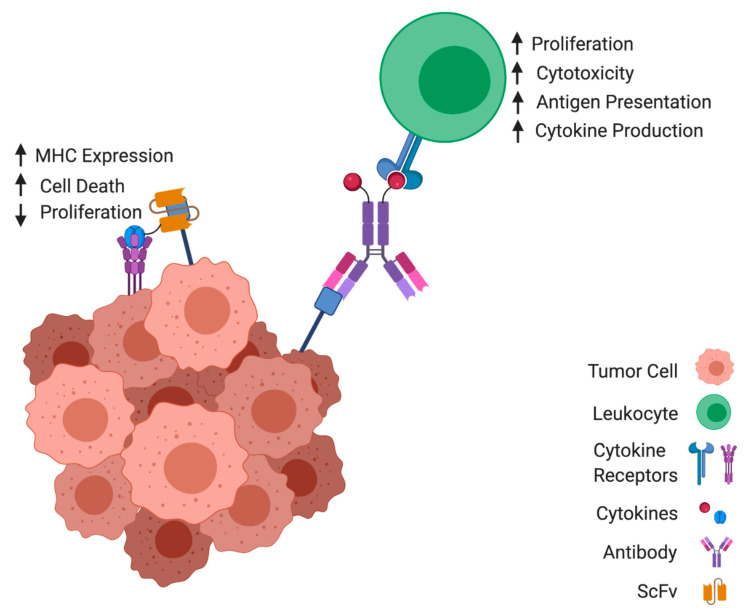
Concept of immunocytokine anti-cancer mechanism.

**Table 1 antibodies-10-00010-t001:** Published clinical trials utilizing immunocytokines.

Target Antigen	Disease	Cytokine Delivered	Phase	Clinical Trial #	Reference
GD2	Neuroblastoma Melanoma Sarcoma Solid childhood tumors	IL-2	I and II	NCT00590824 NCT00082758 NCT03958383 NCT03209869 NCT01334515 NCT00003750 NCT00109863	[195,196,197,198,199,200,201,202,203,204,205]
Tnc A1	Breast carcinoma AML Solid tumors MCC	IL-2	I and II	NCT01131364 NCT01134250 NCT02957032 NCT02054884 NCT03207191	[206,207,208,209]
EpCAM	SCLC Prostate carcinoma Ovarian carcinoma Breast carcinoma Bladder carcinoma Kidney carcinoma Lung carcinoma Solid tumors	IL-2	I and II	NCT00132522 NCT00016237	[210,211,212]
EDB	Melanoma RCC NSCLC Solid tumors Pancreatic carcinoma Colorectal carcinoma DLBCL Glioblastoma Sarcoma Glioma	IL-2 IL-12 TNF	I, II, and III	NCT01058538 NCT02086721 NCT01198522 NCT01253837 NCT02076620 NCT02076646 NCT04471987 NCT02957019 NCT01213732 NCT01055522 NCT04443010 NCT04032964 NCT03420014 NCT03779230 NCT04573192 NCT03705403 NCT02938299 NCT03567889 NCT01253096 NCT00625768 NCT02076633	[159,213,214,215,216,217,218,219,220,221,222]
Histone/DNA structures	NSCLC Solid tumors Pancreatic carcinoma Urogenital carcinoma Bladder carcinoma NHL Kaposi sarcoma Melanoma	IL-2LT IL-12	I and II	NCT04327986 NCT04235777 NCT00879866 NCT01032681 NCT04303117 NCT01973608 NCT01417546 NCT02994953	[223,224,225,226,227]
CEA	Solid tumors	IL-2v	I	NCT02350673 NCT02004106	[228,229]
CD20	B cell lymphoma	IL-2	I and II	NCT02151903 NCT01874288 NCT00720135	[230,231,232]
FAP	Solid tumors RCC Melanoma Pancreatic adenocarcinoma Breast carcinoma HNC Esophageal carcinoma Cervical carcinoma	IL-2v	I and II	NCT03063762 NCT03875079 NCT03193190 NCT02627274 NCT03386721	[233]
PD-1	Solid tumors	IL-2v	I	NCT04303858	

Targets, diseases, delivered cytokines, trial phases, and references that describe clinical trials involving immunocytokines are listed. Antigen targets include GD2: disialoganglioside; Tnc A1: tenascin C A1 domain; EpCAM: epithelial cell adhesion molecule; EDB: fibronectin extra domain B; CEA: carcinoembryonic antigen; and FAP: fibroblast activation protein. Indication abbreviations are as follows. AML: acute myeloid leukemia; MCC: Merkel cell carcinoma; SCLC: squamous cell lung carcinoma; RCC: renal cell carcinoma; NSCLC: non-squamous cell lung carcinoma; DLBCL: diffuse large B cell lymphoma; NHL: non-Hodgkin’s lymphoma; and HNC: head and neck cancer.

## Data Availability

Not applicable.

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
