# Peer review of "Utilizing Immunocytokines for Cancer Therapy"

_2073-4468, 2021, doi:10.3390/antib10010010_

Round 1

Reviewer 1 Report

Revision of

Manuscript ID: antibodies-1091569
Type of manuscript: Review
Title: Utilizing Immunocytokines for Cancer Therapy

The work is well written, it aims to give au up-to-date overview of the immunocytokines for cancer therapy both at preclinical stage and in the clinic. I believe it is worth having this kind of summary to keep scientific translational community informed about the current status of targeted cytokines. I would like to give some suggestions, which are listed below.

#1 IL2 chapter

Beside mentioning it in Table1, I would describe here the anti-PD-1 fusion to IL2-variant (i.e., Roche).

#2 TNF chapter

I would mention here the TNF targeted to EDB, which is reported later in the clinical development section and Table 1 and among references (#193, Spitaleri et al)

#3 Variants chapter

Even if untargeted, given the importance of such biotech deal on IL2-based molecule, I would mention the IL2-PEGylation strategy (Nektar-214). I believe many lessons can be learnt form this trial, also its efficacy when combined with anti-PD1.

#4 General comment

In the immunocytokine field, the size/format of the molecules are important in order to modulate (i) the blood half-life (e.g., IgG-based immunocytokines have longer half lives as compared to antibody-fragments and this may lead to different systemic toxicity) (ii) the penetration in the tumor. Maybe it is worth mentioning this concept in an overview of therapeutic immunocytokines.

Also, if wished, there is another more recent concept of "activation on demand" e.g., 4-1BB agonist (Sci Transl Med, 2019 Jun 12;11(496))

#5 General comment (OPTIONAL)

I would try to improve the figure details (e.g., maybe I would separate (i) the concept of molecule types - examples from literature - (ii) the potential mechanism of actions). I understand if the authors decide to keep one figure only for having just a simple scheme of the concept.

Author Response

Dear Reviewer,

Thank you very much for your recommendations. I have addressed them as follows:

#1 IL2 chapter

Beside mentioning it in Table1, I would describe here the anti-PD-1 fusion to IL2-variant (i.e., Roche). Thank you for this suggestion. I have added a short description of this immunocytokine in the variant section on lines 296-299: The same group also fused this IL-2 variant to a PD-1 targeting antibody to deliver IL-2 directly to effector cells. Administration of the anti-PD-1 IL-2 variant eradicated pancreatic tumors in orthotopic models while co-administration of anti-PD-1 and an irrelevant IL-2 variant immunocytokine did not [127].”

#2 TNF chapter

I would mention here the TNF targeted to EDB, which is reported later in the clinical development section and Table 1 and among references (#193, Spitaleri et al). I have elaborated on the preclinical data of the immunocytokine from the trial described in Spitaleri et al. on lines 209-212: Targeting TNF to EDB delayed tumor growth in teratocarcinoma models compared to untargeted cytokine and the anti-cancer effect was amplified when co-administered with melphalan or an EDB-specific IL-12 immunocytokine [24], [84].” I did not reference Spitaleri et al. here as I describe the results of that study in the clinical trials section.

#3 Variants chapter

Even if untargeted, given the importance of such biotech deal on IL2-based molecule, I would mention the IL2-PEGylation strategy (Nektar-214). I believe many lessons can be learnt form this trial, also its efficacy when combined with anti-PD1. Thank you for this suggestion. The studies for this drug are impressive. It has been referenced in the variants section on lines 312-319: Similarly, specific PEGylation of unmutated IL-2 diminished IL-2Ra binding and biased the molecule towards IL-2Rb. This PEGylated IL-2 exhibited greater tumor retention and circulation half-life, increased infiltrating CD8+ T cells and superior tumor growth inhibition compared to unconjugated IL-2 in multiple murine models [133], [134]. Tumor inhibition was significantly increased when co-administered with checkpoint inhibitors anti-CTLA-4 or anti-PD-1 [133], [135]. This selective IL-2 variant has since progressed to phase II and III clinical trials in combination with anti-PD-1 (nivolumab) for advanced solid tumors, demonstrating a 50% overall response rate in PD-L1+ tumors [136]–[139].”

#4 General comment

In the immunocytokine field, the size/format of the molecules are important in order to modulate (i) the blood half-life (e.g., IgG-based immunocytokines have longer half lives as compared to antibody-fragments and this may lead to different systemic toxicity) (ii) the penetration in the tumor. Maybe it is worth mentioning this concept in an overview of therapeutic immunocytokines. A paragraph has been added to the end of the introduction on lines 80-91: “Similar to monoclonal antibody therapy, the structure and size of the immunocytokine under investigation affects the performance of the drug. Full size antibody formats have increased circulation half-life and engage immune cells through the crystallizable fragment (Fc) portion of the molecule, retaining much of the antibody effector function within the immunocytokine [31], [32]. However, immunocytokines utilizing antibody fragments also increase the half-life of the respective cytokine [24]. Smaller immunocytokine formats have a quicker clearance rate and greater tumor penetration [24], [33]. Additionally, the structure of the cytokine payload can affect efficacy. An IL-12 based immunocytokine demonstrated enhanced tumor retention and growth inhibition when expressed as a single chain variable fragment (scFv) heterodimer connected by a disulfide bridge between the two subunits of IL-12 compared to a monomer consisting of both IL-12 subunits fused by a linker [34]. The intended effect of the immunocytokine and the native conformation of the delivered cytokine should be taken into consideration during engineering.”

Also, if wished, there is another more recent concept of "activation on demand" e.g., 4-1BB agonist (Sci Transl Med, 2019 Jun 12;11(496)) Thank you for this! It is reference 124.

#5 General comment (OPTIONAL)

I would try to improve the figure details (e.g., maybe I would separate (i) the concept of molecule types - examples from literature - (ii) the potential mechanism of actions). I understand if the authors decide to keep one figure only for having just a simple scheme of the concept. I appreciate this recommendation to expand on the detail, however I decided to keep the figure as the original. My goal was to give a brief overview of the concept to an audience potentially unfamiliar with immunocytokines.

Best Wishes,

Erin

Reviewer 2 Report

The authors emphasize the promising prospects of combination therapies involving immunocytokines for cancer therapy.

Considering the substantial expenses for conventional cancer therapy during the last decades, the overall results are rather poor. As for some years the concept to describe cancer as a metabolic disease has been reemerged as a viable base for research on suitable therapies, it may be of interest for the readers, if there are any relevant combinations with metabolic therapies, e.g. ketogenic nutrition. This should be briefly mentioned and relevant research, if reported in literature, included.

Author Response

Dear Reviewer,

Thank you for your comments:

"The authors emphasize the promising prospects of combination therapies involving immunocytokines for cancer therapy.

Considering the substantial expenses for conventional cancer therapy during the last decades, the overall results are rather poor. As for some years the concept to describe cancer as a metabolic disease has been reemerged as a viable base for research on suitable therapies, it may be of interest for the readers, if there are any relevant combinations with metabolic therapies, e.g. ketogenic nutrition. This should be briefly mentioned and relevant research, if reported in literature, included."

Thank you very much for this insight. Cancer metabolism is a rapidly evolving field, as well as metabolism of cellular therapeutics for cancer. Unfortunately, I was unable to find metabolic research in the context of immunocytokine therapy and therefore did not include this concept in the review.

Best Wishes,

Erin

Reviewer 3 Report

The review by Runbeck et al focuses on the immunocytokines for targeted cancer therapy. The manuscript is well written and scientifically sound. What I miss is a subchapter focusing on engineering of immunocytokines-what are the strategies to generate immunocytokines? What are they structural components and what for? Are there any modifications of mAb or cytokines in fusion to improve their action? Elaborating on these points in the initial parts of review will facilitate understanding of specific immunocytokines action.

Author Response

Dear Reviewer,

Thank you for your recommendations for the manuscript. I have addressed them as follows:

"The review by Runbeck et al focuses on the immunocytokines for targeted cancer therapy. The manuscript is well written and scientifically sound. What I miss is a subchapter focusing on engineering of immunocytokines-what are the strategies to generate immunocytokines? What are they structural components and what for? Are there any modifications of mAb or cytokines in fusion to improve their action? Elaborating on these points in the initial parts of review will facilitate understanding of specific immunocytokines action."

I included a sentence as a brief definition of immunocytokines on line 68-69: “They are composed of a targeting-antibody moiety, an amino acid linker and a cytokine payload.”

I also provided a new paragraph at the end of the introduction, expanding on the format and structure of immunocytokines and how that changes their properties (line 80-91): “Similar to monoclonal antibody therapy, the structure and size of the immunocytokine under investigation affects the performance of the drug. Full size antibody formats have increased circulation half-life and engage immune cells through the crystallizable fragment (Fc) portion of the molecule, retaining much of the antibody effector function within the immunocytokine [31], [32]. However, immunocytokines utilizing antibody fragments also increase the half-life of the respective cytokine [24]. Smaller immunocytokine formats have a quicker clearance rate and greater tumor penetration [24], [33]. Additionally, the structure of the cytokine payload can affect efficacy. An IL-12 based immunocytokine demonstrated enhanced tumor retention and growth inhibition when expressed as a single chain variable fragment (scFv) heterodimer connected by a disulfide bridge between the two subunits of IL-12 compared to a monomer consisting of both IL-12 subunits fused by a linker [34]. The intended effect of the immunocytokine and the native conformation of the delivered cytokine should be taken into consideration during engineering.”

Additionally, I previously mentioned some modifications of immunocytokine structure that can improve their efficacy. Some examples are on lines 231-233: “Variability has also been observed with different linker lengths and sequences attaching IL-12 and IL-2, with specific linkers enhancing tumor localization and retention [91], [92].” And lines 306-308: “An alternative structure design of cytokine conjugation to the light chain instead of the heavy chain of a full length antibody increased the half-life and therapeutic effect through hinderance of the IL-2Rb binding portion of IL-2 [130].”

Best Wishes,

Erin